# How Do Enveloped Viruses Exploit the Secretory Proprotein Convertases to Regulate Infectivity and Spread?

**DOI:** 10.3390/v13071229

**Published:** 2021-06-25

**Authors:** Nabil G. Seidah, Antonella Pasquato, Ursula Andréo

**Affiliations:** 1Laboratory of Biochemical Neuroendocrinology Montreal Clinical Research Institute, University of Montreal, Montreal, QC H2W1R7, Canada; Ursula.Andreo@ircm.qc.ca; 2Antonella Pasquato, Department of Industrial Engineering, University of Padova, Via Marzolo 9, 35131 Padova, Italy; Antonella.pasquato@unipd.it

**Keywords:** enveloped virus, proprotein convertases, Furin, SKI-1/S1P, PCSK9, SARS-CoV-2, COVID-19, pandemic

## Abstract

Inhibition of the binding of enveloped viruses surface glycoproteins to host cell receptor(s) is a major target of vaccines and constitutes an efficient strategy to block viral entry and infection of various host cells and tissues. Cellular entry usually requires the fusion of the viral envelope with host plasma membranes. Such entry mechanism is often preceded by “priming” and/or “activation” steps requiring limited proteolysis of the viral surface glycoprotein to expose a fusogenic domain for efficient membrane juxtapositions. The 9-membered family of Proprotein Convertases related to Subtilisin/Kexin (PCSK) serine proteases (PC1, PC2, Furin, PC4, PC5, PACE4, PC7, SKI-1/S1P, and PCSK9) participate in post-translational cleavages and/or regulation of multiple secretory proteins. The type-I membrane-bound Furin and SKI-1/S1P are the major convertases responsible for the processing of surface glycoproteins of enveloped viruses. **Stefan Kunz** has considerably contributed to define the role of SKI-1/S1P in the activation of arenaviruses causing hemorrhagic fever. Furin was recently implicated in the activation of the spike S-protein of SARS-CoV-2 and Furin-inhibitors are being tested as antivirals in COVID-19. Other members of the PCSK-family are also implicated in some viral infections, such as PCSK9 in Dengue. Herein, we summarize the various functions of the PCSKs and present arguments whereby their inhibition could represent a powerful arsenal to limit viral infections causing the present and future pandemics.

## 1. Introduction

Infectious diseases that threatened the life of humans since at least the Neolithic times, are believed to have started around 12,000 years ago, when roaming human hunter-gatherers became sedentary and settled into small camps and villages to domesticate animals and cultivate crops [1,2]. The domestication of animals and their proximity to humans favored the transmission of animal-human (zoonotic) diseases. The latter implicated various infectious microorganisms, such as bacteria, fungi, parasites, and viruses. Some of these pathogens caused widespread disease worldwide with “pandemic proportions” resulting in the death of a significant fraction of the human population.

The earliest recorded pandemic disease, called Antonine Plague (A.D. 165–180), resulted from infectious smallpox or measles virus and led to ~5 million (M) deaths throughout the Roman empire (~10% of the empire’s population). The next three big pandemics were caused by deadly bacterial (Yersinia pestis) infections. The first was the Justinian’s Plague (A.D. 541–542), which resulted in 5–10 M fatalities, was traced to China and northeast India. The bacterial infection was transmitted via land and sea trade routes to Egypt where it entered the Byzantine Empire through Mediterranean ports. The second was also caused by the same gram-negative bacteria transmitted from rats via infected fleas to humans and was at the origin of the bubonic/pneumonic plague (black death; A.D. 1347–1351), which resulted in the highest number of fatalities ever (200 M). The third was a major bubonic plague pandemic that began in Yunnan, China, in 1855, spread to all inhabited continents leading to 12–15 M+ deaths in India and China, with about 10 M killed in India alone. The Smallpox virus (Variola, A.D. 1520) killed more than 56 M people over the years resulting in the decimation of 90% of Native Americans and 400,000 death/year in Europe in the 19th century. This virus must have circulated for a long time since signs of smallpox have been found in Egyptian mummies, including Ramses V, who died in 1157 B.C. The deadliest pandemic virus in the 20th century (A.D. 1918–1919) was traced to an H1N1 variant of the flu virus (Spanish Flu) and resulted in 50 M+ death worldwide close to the end of the first world war. This virus spread between the soldiers in the war trenches in Europe and was transmitted to the general population when they came back home. Such a deadly pandemic never went away and is still causing problems today. Finally, another persisting pandemic that has hit the world starting in 1981 is due to HIV retroviral infections causing AIDS, leading to more than 35 M+ death worldwide. The latest ongoing viral-related pandemic that started in December 2019 causes a coronavirus-associated disease (called COVID-19) due to SARS-CoV-2 infections [3], which so far have resulted in more than 2.8 M+ death worldwide and against which we now have access to various efficient vaccine formulations [4,5,6]. 

From the above summary, it is apparent that a substantial proportion of the deadly newly emerging and re-emerging diseases in the past century have been of viral origin [6]. While not all viruses have envelopes, e.g., enteric RNA viruses [7], the majority (RNA & DNA viruses and retroviruses) have a viral envelope that protects the genetic material in their life cycle when traveling between host cells. The envelopes are typically derived from portions of the host cell membranes (phospholipids and proteins) but include some viral glycoproteins. Glycoproteins on the surface of the envelope serve to identify and bind to receptor sites on the host’s membrane. The viral envelope then fuses with the host’s cell membrane [8], allowing the capsid and viral genome to enter the host. Another mechanism of virus spread requiring fusion is the formation of syncytia through cell-to-cell fusion. An infected cell harboring the glycoprotein on the plasma membrane can fuse with an adjacent cell forming polynucleated infected cells called syncytia. This mechanism allows the virus to spread while limiting the release of the virus in the extracellular milieu. 

The proprotein convertases (PCs; genes *PCSKs*) constitute a family of nine secretory serine proteases that regulate various processes in both health and disease states [9]. Seven basic amino acid (aa)-specific convertases (PC1, PC2, Furin, PC4, PC5, PACE4, and PC7) cleave precursor proteins at single or paired basic amino acids after the general motif (K/R)-X_n_-(K/R)↓, where Xn = 0, 2, 4 or 6 spacer X residues [9,10]. The 8th member (SKI-1/S1P) cleaves precursor proteins at R-X-Aliphatic-Z↓, where X is any residue except Pro and Cys, and Z is any aa except Val, Pro, Cys, or Glu [11,12,13,14,15]. The last member, PCSK9, cleaves itself once, but its inhibitory prodomain remains non-covalently attached rendering it inactive as a protease *in trans*, but rather it can act as a chaperone escorting some surface receptors (e.g., LDLR and MHC-class I receptor) toward lysosomes for degradation [16,17,18,19]. 

Through proteolysis, PCs are responsible for the activation and/or inactivation of many secretory precursor proteins, including viral surface glycoproteins [9,10,20] (Table 1). 

Because of their critical functions, PCs, especially the ubiquitously expressed Furin [22] and SKI-1/S1P [11] are implicated in many viral infections via specific cleavages of envelope glycoproteins, a condition that allows not only the fusion of the viral lipid envelope with host cell membranes [9] but also for cell-to-cell fusion forming syncytia of certain viruses leading to important cytopathogenic effects [23,24].

In this review, we will summarize our knowledge of the role of enzymatic cleavage by proprotein convertases and type-II transmembrane serine proteases required to facilitate virus entry through fusion as well as an innate antiviral strategy that the host deploys to counteract this step. A number of excellent reviews have appeared on the importance of cleavage of the surface glycoproteins of enveloped viruses in productive infections [21,25]. Herein, we will mostly concentrate on the roles of the proprotein convertases [9] in this process and emphasize the seminal contributions of the group of **Stefan Kunz** in the analysis of the function of SKI-1/S1P in the activation of arenaviruses causing hemorrhagic fever. 

## 2. Proprotein Convertases and Enveloped Viruses

### 2.1. Furin in Viral Infections and Pathogenicity

Furin is the third and best characterized member of the PCSK family of secretory convertases. It cleaves at basic amino acid motifs and recognizes a prototypical sequence R-X-(K/R)-R↓ [10]. The presence of this Furin(-like) motif in viral envelope glycoproteins constitutes a way to activate the fusion-dependant entry of several viruses such as HIV gp160, influenza hemagglutinin (Table 2), and some coronaviruses spike proteins [26,27].

In 1992, Furin was identified to be the cellular protease cleaving hemagglutinin (HA) of fowl plague/influenza virus (FPV; H7N1 A/FPV/Rostock/34), as well as the HIV glycoprotein gp160 into gp120 and gp41 [28,29,30]. The presence of a Furin cleavage sequence can shift the pathogenicity of certain viruses from non-virulent to virulent. For instance, Newcastle disease virus (NDV) fusion protein contains only a monobasic cleavage site, while the virulent NDV fusion protein possesses a polybasic sequence of amino acids that can be cleaved by Furin [31]. Similarly, the acquisition of a Furin cleavage site has been linked to the high pathogenicity of avian influenza [32], especially for the Hong Kong variant (H5N1 A/HK/97) [33,34] that has an insertion of RERR between the NTPQ and the Furin-site RKKR↓GL in HA (Table 2), resulting in a very efficient cleavage by more than one convertase, such as Furin, PC5 and PC7 [35]. Other pathogenic influenza virus do not possesses a Furin cleavage site and rely on trypsin-like proteases, e.g., HAT and TMPRSS2 [36]. The expression of these enzymes is restricted to the upper respiratory tract [37], while Furin is ubiquitously expressed, and the acquisition of a Furin(-like) site enhances the cellular tropism of influenza. However, the presence of a polybasic Furin(-like) cleavage site is not the only determining factor for cleavage by Furin, as the amino-acids adjacent to the Furin(-like) sequence are important too, as described in Table 2. Furthermore, the presence of oligosaccharides near the cleavage site can prevent processing [38], unless it is compensated by an increased number of basic residues as described in the non-pathogenic variant of influenza hemagglutinin A of H5N2 [39].

Aside from glycoproteins from NDV and influenza hemagglutinin, HIV glycoprotein, as well as other retrovirus glycoproteins such as Rous sarcoma virus and murine leukemia virus are cleaved by Furin. Interestingly, these glycoproteins form trimers, and some viruses assemble with un-cleaved glycoproteins that remain either inactive or can be cleaved at the plasma membrane. The Furin-specific cleavage of gp160 in HIV-1 has been described as dispensable as other proprotein convertases [40] could substitute for Furin in some tissues, and plasmin could activate the gp160 at the plasma membrane. However, these hypotheses remain to be validated in vivo in humans.

Human metapneumovirus (hMPV) is a paramyxovirus responsible for acute respiratory tract infections which can result in hospitalization of both children and adults. We showed that blocking the activity of the protease-activated receptor 1 (PAR1), a G-coupled receptor, induced by inflammation, protects against hMPV infections [41]. Furthermore, Furin activates the envelope F-protein of hMPV by cleavage at NP**R**QS**R**_102_↓F**V** (Table 2, where bold residues emphasise the critical P1 and P4 positions) [41]. Unexpectedly, PAR1 itself potently inhibits cellular Furin activity [42]. Indeed, PAR1 exhibits in its second luminal loop a Furin(-like) motif (P**R**SFLL**R**_46_-NP) with a P1 and P6 Arg and is cleaved by PC5A and PACE4 but not by Furin. The presence of an Asn_47_ at the P1′ site rather made it a potent Furin-inhibitor that binds its catalytic subunit and sequesters the [PAR1-Furin] complex in the *trans*-Golgi network (TGN), thereby inactivating Furin and preventing PAR1 from reaching the cell surface [42]. The overexpression of PAR1 has been described in the brain upon neuroinflammation and in patients with neurocognitive disorders associated with HIV infection (HAND) [42]. Interestingly, while PAR1 binds and inhibits Furin, PAR2 is cleaved by Furin at the motif **R**SSKG**R**_36_↓S**L**, which also exhibits a P1 and P6 Arg but a favorable Ser at P1′ and Leu at P2′ (Table 2) [38]. Recently, another strategy of viral host-cell defense has been identified. Guanylate-binding proteins (GBPs) are interferon stimulated genes. Upon HIV infection, the cytosolic GBP2 and 5 expressions are induced, and these proteins can bind the cytosolic tail of Furin and prevent its trafficking beyond the *cis/medial* Golgi, thus keeping it in an inactive state bound to its inhibitory prodomain. This effectively inhibits the Furin processing of gp160 into gp120 and gp41 [25,43]. Thus, while PAR1 is the first secretory endogenous natural inhibitor of Furin that blocks its enzymatic activity and prevents its exit from the TGN, the cytosolic GBP2,5 also inhibits Furin activity by blocking it in the *cis/medial* Golgi in an inactive state.

Another family of viruses requiring Furin is the *filoviridae*. Marburg and Ebola are enveloped single negative strand RNA viruses of the *filoviridae* family. These viruses are highly pathogenic in humans causing hemorrhagic fever with a very high mortality rate. The glycoprotein of Marburg and Ebola (GPs) are cleaved by Furin into GP1 containing the receptor binding domain (RBD) and membrane-bound GP2 [44]. Pathogenic viruses of the family possess a Furin(-like) cleavage site, but experiments design to establish the role of Furin in pathogenicity indicated that Furin cleavage is not required for replication in cell culture [45] and disease severity in non-human primates [46]. Recent work identified a new Furin site in Ebola GP that requires N-glycosylation to be processed [47]. Interestingly, MARCH8 a member of the Membrane-associated RING-CH-type 8 (MARCH8) that has been described to have broad antiviral activity against several viruses, can inhibit Furin by forming a complex with the GP and Furin thereby sequestering it in the Golgi, and preventing the maturation of the viral particles [48]. This is the third example of endogenous Furin-inhibitors that sequester the enzyme in a subcellular Golgi compartment where it remains inactive. 

The Flaviviruses envelope also requires Furin to get activated. The first evidence was reported for the Tick Borne Encephalitis virus. Flavivirus are enveloped positive strand RNA viruses. Their surface protein is composed of a heterodimer formed by prM and E [49]. The immature prM is cleaved by Furin at GS**R**T**RR**_205_↓S**V** before the viral particles are released from the cell. The cleavage allows fusion and is required for infectivity. An acidic pH is required to promote a change of conformation of the heterodimer to facilitate Furin cleavage. While, the cleavage of prM into M is inefficient especially for Dengue virus (**H**R**R**E**KR**_205_↓S**V**) (Table 2), likely due to the presence of a Glu at P3 [21], Furin has been confirmed to be required for infectivity of Dengue virus in vitro by using LoVo cells that are Furin deficient [50]. Additionally, the role of Furin in the antibody-dependent-enhancement (ADE) of Dengue virus infection is intriguing. ADE is the mechanism by which the immunisation against Dengue could exacerbate the pathogenicity of Dengue in the context of a second infection. This has been explained by the fact that the binding of antibodies to an immature particle could enhance its infectivity. Interestingly, while Furin inhibition has been described to block anti-Envelope dependant enhancement [51], antibodies against prM (uncleaved by Furin) allow tissue entry of immature particles [52]. Recently, broadly neutralizing monoclonal antibodies were reported to protect against multiple tick-borne flaviviruses (TBFVs) [53], opening the way to the design of vaccines and antibody therapeutics against clinically relevant TBFVs. 

Chikungunya virus (CHIKV) is a mosquito-transmitted α-virus that causes in humans an acute infection characterized by polyarthralgia, fever, myalgia, and headache. Since 2005 this virus has been responsible for an epidemic outbreak of unprecedented magnitude. By analogy with other α-viruses, it is thought that cellular proteases can process the viral precursor protein E3E2 to produce the receptor binding E2 protein that associates as a heterodimer with E1. Destabilization of the heterodimer by exposure to low pH allows viral fusion and infection. We demonstrated that membrane-bound Furin but also PC5B can process E3E2 from African CHIKV strains at the H**R**Q**RR**_642_↓ST site, whereas a CHIKV strain of Asian origin is cleaved at R**R**Q**RR**_642_↓SI (Table 2) by membranous and soluble Furin, PC5A, PC5B, and PACE4 but not by PC7 or SKI-1/S1P [54]. This cleavage was also observed in CHIKV-infected cells and could be blocked by Furin inhibitor decanoyl-RVKR-chloromethyl ketone. This inhibitor was compared with chloroquine for its ability to inhibit CHIKV spreading in myoblast cell cultures, a cell-type previously described as a natural target of this virus [54]. We observed an additive effect of dec-RVKR-cmk and chloroquine, supporting the concept that these two drugs act by essentially distinct mechanisms. The combinatory action of chloroquine and dec-RVKR-cmk led to almost total suppression of viral spread and yield [54]. 

Furin cleavage of viral glycoproteins of the *Herpesviridae* family have been reported for Herpes simplex virus 1 and 2 and Varicella Zoster (Table 1), contributing to the pathogenesis of the later in vivo [55,56]. Finally, while Furin cleaves many glycoproteins, its activity is required for other viral proteins such as for Hepatitis B virus (HBV) core antigen. For reviews see [21,25]. 

HIV-1 encodes four accessory gene products—Vpr, Vif, Vpu, and Nef—which are thought to collectively manipulate host cell biology to promote viral replication, persistence, and immune escape [57]. Increasing evidence suggests that extracellular Vpr could contribute to HIV pathogenesis through its effect on bystander cells. Soluble forms of Vpr have been detected in the sera and cerebrospinal fluids of HIV-1-infected patients, and in vitro studies have implicated extracellular Vpr as an effector of cellular responses, including G2 arrest, apoptosis, and induction of cytokines and chemokines production, presumably through its ability to transduce into multiple cell types. Thus, Vpr is a true virulence factor and is a potential and promising target in different strategies aiming to fight infected cells including latently HIV-infected cells [58]. While Furin is clearly implicated in the processing activation of multiple viral surface glycoproteins, including gp160 of HIV-1, other cell-surface associated convertases such as PC5A and PACE4 [9,59] seem to negatively regulate the activity of the HIV-1 by cleavage-inactivation of secreted Vpr at **R**Q**RR**_88_↓ [60]. PC-mediated processing of extracellular Vpr results in a truncated Vpr product that was defective for the induction of cell cycle arrest and apoptosis when expressed in human cells [60]. Thus, inhibitors of PC5/PACE4 may enhance HIV-1 virulence, and Furin-specific inhibitors would restrict HIV-1 infectivity, without affecting Vpr inactivation by PC5A/PACE4. 

### 2.2. Coronavirus Infections, Including SARS-CoV-2

There are seven known human coronaviruses (CoV), which are enveloped positive-stranded RNA viruses belonging to the order *Nidovirales* and are mostly responsible for upper respiratory tract infections. All these coronaviruses exhibit a “crown-like” structure composed of a trimeric spike (S) protein. The S-protein is a ~180–200 kDa type I transmembrane protein, with the N-terminus facing the extracellular space and anchored to the viral membrane via its transmembrane domain followed by a C-terminal short tail facing the cytosol [61]. During infection, the trimetric S-protein is first processed at the “priming site” S1/S2 by host cell proteases, separating the N-terminal S1 domain that binds a specific host cell receptor(s) via its receptor binding domain (RBD), and the C-terminal membrane-bound S2 domain involved in viral entry through fusion at the plasma membrane after cleavage. S1 and S2 have been described to remain non-covalently bound therefore protecting the fusion peptide from being exposed too early before the membranes can fuse. Four coronaviruses are now endemic in the human population and cause mild disease, including the α-coronaviruses HCoV-229E and HCoV-NL63 and β-coronaviruses HCoV-OC43 and HCoV-HKU1 [24]. Interestingly, the insertion of a more favourable S1/S2 Furin(-like) site of the neuro-invasive HCoV-OC43 (R**R**S**RR**_758_↓A**I**) resulted in the establishment of less pathogenic but more persistent viral infections in the brain [26]. 

In the last 18 years three new highly pathogenic human β-coronaviruses appeared, SARS-CoV-1 in 2002–2003, MERS-CoV in 2013, and SARS-CoV-2 (Table 2) at the end of 2019. The first two were associated with severe human pathologies, such as pneumonia and bronchiolitis, and even meningitis in more vulnerable populations [62]. However, SARS-CoV-1 was rapidly contained and MERS-CoV only spread from dromedary to human and no human to human transmissions were reported. The RBDs of the S1-subunits of SARS-CoV-1 and MERS-CoV recognise angiotensin-converting enzyme 2 (ACE2) [63] and dipeptidylpeptidyl 4 (CD26/DPP4) [64] as their entry receptors, respectively. While the single basic Arg at the S1/S2 site of SARS-CoV-1 (TVSLL**R**_667_-ST) does not contain a Furin(-like) site, that of MERS-CoV does (TP**R**SV**R**_751_↓S**V**). Indeed, the priming of SARS-CoV-1 in the lung is thought to be performed by elastase [65,66], whereas MERS-CoV processing at S1/S2 is thought to be mostly performed by Furin and possibly by TMPRSS2, a type-II membrane-bound serine protease [67,68]. The S2-product generated following S1/S2 cleavage contains a second proteolytic site (called S2′ cleavage site), which when cleaved would generate an S2′-fragment that starts with a fusion peptide (FP) followed by two heptad-repeat domains preceding the transmembrane domain (TM) and cytosolic tail. Cleavage at the S2′ site triggers membrane fusion and is essential for efficient viral infection. The S2′ cleavage sites of SARS-CoV-1 (PT**KR**_797_↓SF) and MERS-CoV (**R**SA**R**_887_↓SA) suggest that both could be cleaved by a Furin(-like) enzyme, but the single Arg-specific enzyme TMPRSS2 has solely been proposed to cleave both spike glycoproteins at the S2′ site [69]. 

The Furin(-like) S2′ cleavage site at **KR**_797_↓S**F** with P1 and P2 basic residues and a P2′ hydrophobic Phe [9], is identical between the SARS-CoV-1 and SARS-CoV-2 (Figure 1). In the MERS-CoV and OC43-CoV it is replaced by the less favourable Furin(-like) site **R**XX**R**↓SA, with P1 and P4 basic residues, and an Ala (not hydrophobic) at P2′. However, in other less pathogenic circulating human coronaviruses, the S2′ cleavage site only exhibits a Furin-unfavourable monobasic **R**↓S sequence [70] with no basic residues at either P2 and/or P4. Even though processing at the S2′ site in the spike glycoprotein of SARS-CoV-2 is thought to be key in the activation of the S-protein, leading to cell-fusion and entry, multiple protease(s) might be involved in S-cleavage at different sites and subcellular compartments [71]. The ability of the Arg/Lys-specific TMPRSS2 to directly cleave at the S2′ site was inferred from the viral entry blockade by the relatively non-specific TMPRSS2 inhibitor Camostat [72,73]. We recently demonstrated that Furin is capable of performing both S1/S2 (P**R**RA**R**_685_↓S**V**) (Table 2) and S2′ (KPS**KR**_815_↓S**F**) cleavages generating an N-terminal S1 subunit (with an RBD domain) and a C-terminal membrane bound fusogenic S2′-fragment in the presence of the Spike-glycoprotein receptor ACE2 [74]. Interestingly, binding of the RBD of the S-protein of SARS-CoV-2 to ACE2 was demonstrated to exert a conformational change that allosterically enhances the exposure of the S1/S2 site to Furin cleavage [75]. Whether it also enhances the exposure of the S2′ site is not clear, but our results clearly show that Furin cleavage at the S2′ site is enhanced in the presence of ACE2 [74]. Our data also showed that non-peptide small molecule BOS-inhibitors of Furin(-like) enzymes block the processing of the S-protein at both S1/S2 and S2′ sites and result in a significant reduction of cell-to-cell fusion. In the presence of overexpressed TMPRSS2 together with the S-protein, the latter was cleaved into an apparently lower sized fragment than S1 (called S1′) [74] released in the cell culture media, which turned out to be an authentic S1 product lacking O-glycosylation as it is generated in the endoplasmic reticulum (ER) and secreted by an unconventional pathway (*submitted*). Furthermore, overexpressed TMPRSS2 generated two C-terminal membrane bound fragments S2a and S2b, which are S2 and S2′ products lacking O-glycosylation. These products may be artificially produced in the ER upon overexpression of TMPRSS2 with the S-glycoprotein. However, in co-culture conditions in which ACE2 and TMPRSS2 are expressed in acceptor cells and S-protein in donor cells, we concluded that both Furin (in the TGN) and TMPRSS2 (at the cell surface) cleave S-protein into S1/S2 and S2′ in the presence of ACE2 (*submitted*), but that in addition TMPRSS2 can also process and shed ACE2 into a soluble form thereby modulating viral entry and infection. In agreement, an inhibitor cocktail that combines a Furin inhibitor (BOS) with a TMPRSS2 inhibitor (Camostat) can reduce by 99% SARS-CoV-2 infectivity of lung-derived Calu-3 cells [74]. It should be noted that coronaviruses may enter the cells through fusion at the plasma membrane or the following endocytosis into endosomes [76]. The versatility of strategies to penetrate the cell allow coronaviruses to depend less on the availability of the different proteases in specific cell type and in cellular compartments. Furthermore, the activation process of SARS-CoV-2 may be more complex, as other proteases may also participate in processing the S-protein at multiple sites in various tissues, including cathepsins [71], HAT [77], TMPRSS11D, and TMPRSS13 [78]. 

Interestingly, we previously showed that chloroquine, initially used as an antimalarial, can reduce viral infection and spread of both the Chikungunya virus [54] and SARS-CoV-1 [79]. In the latter case, chloroquine was also shown to affect the N-glycosylation pattern of ACE2, the receptor of SARS-CoV-1 [79] and SARS-CoV-2 [27]. Interfering with terminal glycosylation of ACE2 may negatively influence the virus-receptor binding and abrogate infection, with further ramifications by the elevation of TGN/vesicular pH, resulting in the inhibition of infection and spread of SARS CoV-1 at clinically admissible concentrations. This may also be applicable for SARS-CoV-2, but the protective role of the less toxic hydroxychloroquine together with the antibiotic azithromycin against COVID-19 at the early infection stages remains controversial and begs for unbiased clinical trials [80,81]. 

Worldwide distribution of various SARS-CoV-2 vaccines conceived to mount an immune response against the S protein, therefore, blocking accessibility of the S-protein to ACE2 (https://www.raps.org/news-and-articles/news-articles/2020/3/covid-19-vaccine-tracker, accessed on 17 June 2021) represents an efficient strategy to prevent SARS-CoV-2 infections end to eliminate the burden of the pandemic on a global scale. Time is needed to determine if these vaccines will confer long-lasting protection. Furthermore, since April 2020 new variants of SARS-CoV-2 emerged, some of which are becoming prominent in Europe as well as North and South America. The emergence of such variants indicated an enhanced SARS-CoV-2 fitness and are thought to have increased transmissibility. Variants in the United Kingdom (B.1.1.7; α-variant) [82], South Africa (B.1.351; β-variant) [83], Brazil, and Japan (B.1.1.248; γ-variant) and India (B.1.167; δ-variant) [84] present multiple substitutions in the spike protein, some of which are common among different variants. These include critical mutations in the ACE2 binding site (e.g., N501Y and E484K) within the RBD (Table 3), resulting in enhanced binding affinity of S-protein to ACE2 by 2.5-fold for N501Y and 13.5-fold for [N501Y + E484K] [85]. Interestingly, the α- and δ-variant exhibit mutations in the P5 position of the Furin cleavage site P681H and P681R, respectively, which may increase processing efficacy by Furin(-like) enzymes at the S1/S2 site and enhance viral entry. 

Accordingly, it is not surprising that these variants are associated with a reduced efficacy of certain vaccines against SARS-CoV-2 infection but fortunately remain protective against severe disease and death [86,87]. Amazingly, screening various artificial mutants in vitro showed that some mutant-combinations could result in a very worrisome 600-fold higher binding affinity of S-protein to ACE2 [85]. Thus, B.1.351 and similar spike mutations present new challenges for antibody therapy and threaten the protective efficacy of current vaccines.

While the protective effect of the vaccination on the whole world population remains incomplete, additional effective antiviral drugs that block viral entry in target cells (Figure 1) are still needed and could help in the early treatment of the disease. Ultimately, in the case of new emerging coronavirus pandemics [6,88], the availability of such treatments would constitute a powerful antiviral arsenal.

### 2.3. PCSK9 and Viral Infections

The secreted protein PCSK9 is the 9th member of the PCSKs family that autocatalytically cleaves its prodomain in the ER allowing its exit from this compartment and its secretion from cells [16], as a protease-inactive [prodomain-PCSK9] complex [89,90]. Its high expression in the liver and the presence of its gene on chromosome 1p32 [16], led to the genetic association of gain-of-function (GOF) variants of *PCSK9* with an autosomal dominant form of hypercholesterolemia [91]. This strongly suggested that PCSK9 may play a critical role in low density lipoprotein- cholesterol (LDLc) regulation. Indeed, while GOF variants were associated with high levels LDLc [91], the reverse was true for loss-of-function (LOF) variants [92]. Mechanistically, circulating PCSK9 was shown to reduce the levels of liver LDL-receptor (LDLR) protein by sorting the PCSK9-LDLR complex to lysosomes for degradation, implicating an interaction of the PCSK9 catalytic domain with the EGF-A domain of the LDLR [89,93,94]. 

In a first study, we showed that PCSK9 can reduce HCV infection of HuH7 cells using the cell culture-derived HCV clone JFH1 (genotype 2a). We demonstrated that the effect was due to the PCSK9-induced decrease of the LDLR protein levels on the surface of HuH7 cells. Also, HCV upregulates the expression of LDLR in vitro and in the chronically infected liver [95], likely due to an upregulation of the LDLR promotor activity through SREBPs and a decreased expression of PCSK9 [96]. Befittingly, Alirocumab, a therapeutic human monoclonal antibody (mAb) to PCSK9 used to treat hypercholesterolemia through an increase of cell surface LDLR failed to reduce HCV entry and infectivity in hepatocyte in vitro [97]. 

Additionally, the role of PCSK9 during natural HCV infection is not fully understood. We showed that HCV genotype 3 (G3) infected patients with high viral load have low levels of PCSK9 and LDLc compared to HCV genotype 1 (G1) infected patients, suggesting that the particular interplay between HCV and lipid metabolism and PCSK9 might be genotype specific [98]. In contrast, in a retrospective study, PCSK9 levels were found to be elevated especially in chronic genotype 2 (G2) HCV patients with or without hepatocellular carcinoma (HCC), but the increased PCSK9 levels were not significantly associated with changes in LDLc [99]. Overall, the data reported illustrate the complexity of the relation between HCV and lipid metabolism and more work would be required to better understand the role of PCSK9 in HCV infection. 

Dengue virus (DENV), a single positive-stranded RNA virus of the family Flaviviridae, is transmitted to humans by the urban-adapted *Aedes* mosquitoes. It is estimated that 400 M+ individuals are infected with 1 of the 4 types of DENV [100]. The DENV surface glycoprotein prM-E complex needs cleavage by Furin(-like) enzymes to remove the prodomain of the envelope glycoprotein prM and allow the formation of the activated M-E complex and acquisition of infectivity. However, unlike cleavage of the prM of other flaviviruses, cleavage of DENV prM is incomplete in many cell lines, likely due to the negative influence of the aspartic acid (D) at the P3 position of the processing site HR**R**D**KR**↓S**V** at the pr-M junction (Table 2) [101]. Therefore, it is not surprising that inhibitors of Furin(-like) enzymes, while significantly reducing viral entry, did not completely block viral infections [102] (*unpublished results*). 

Very recently, we showed that DENV infection reduces the antiviral response of the host hepatocytes. Thus, DENV infection induces expression of PCSK9, thereby reducing cell surface levels of LDLR and LDLc uptake resulting in enhanced *de novo* cholesterol synthesis and its enrichment in the ER. In turn, high levels of ER cholesterol suppressed the phosphorylation and activation of the ER-resident stimulator of interferon (IFN) gene (STING), leading to reduction of type I interferon (IFN) signaling through antiviral IFN-stimulated genes (ISGs) [103]. This was supported by the detection of elevated plasma PCSK9 levels in patients infected with DENV resulting in high viremia and increased severity of plasma leakage. This unexpected role of PCSK9 in dengue pathogenesis, led us to test the effect of inhibition of PCSK9 function by the mAb Alirocumab. Befittingly, this treatment resulted in higher LDLR levels and lower viremia. Our data suggested that PCSK9 inhibitors could be a suitable host-directed treatment for patients with dengue [103], possibly in combination with Furin(-like) inhibitors.

### 2.4. Implications of SKI-1/S1P in Viral Infections

The 8th member of the proprotein convertase family was discovered in 1998/1999 as a type-I membrane-bound protease called site-1 protease (S1P) [104] or alternatively subtilisin-kexin isozyme-1 (SKI-1) [11]. Since then, it is commonly named SKI-1/S1P, and its gene is given the name Membrane-Bound Transcription-factor Site-1 Protease (*MBTPS1*) [9]. Extensive studies of its cleavage specificity revealed that it recognizes the general motif **R**-X-**Aliphatic**-Z↓, where X is any residue except Pro and Cys, and Z is any aa (best Leu) except Val, Pro, Cys, or Glu [11,12,13,14,15] (Table 4). 

Different from the basic-aa specific convertases this enzyme undergoes in the ER two autocatalytic zymogen processing events in the middle of the prodomain **R**K**V**F_133_↓**R**S**L**K_137_↓YA (sites B and B’; Table 4) [13,105], allowing it to exit the ER and reach its final destination, the *cis*/*medial* Golgi, where it is further cleaved at the C-terminus of the prodomain at **R**R**A**S_166_↓LS and **R**R**L**L_186_↓RA (sites C’ and C; Table 4) [11,12]. This results in the maximal activation of the protease, which would then act *in trans* to cleave its substrates usually in the *cis*/*medial* Golgi where it is expected to be fully active. 

SKI-1/S1P was first identified in our group because of its ability to cleave in the *cis/medial* Golgi brain derived neurotrophic factor (BDNF) at **R**G**L**T_57_↓SL, thereby preceding the Furin cleavage in the TGN at **R**V**RR**_128_↓HS to release mature BDNF [11]. The enzyme was then linked to the activation of several transcription factors. Among others, the sterol regulatory element binding proteins 1 and 2 (SREBP1,2; implicated in sterol and fatty acid synthesis), the type-II membrane-bound ER-stress factor ATF6, and other type-II membrane-bound substrates including at least six CREB-like basic leucine zipper transcription factors (Table 4). Other substrates identified revealed a role of SKI-1/S1P in the regulation of the kinase FAM20C that phosphorylates luminal serine residues in the secretory pathway [106], and in the activation of the α/β-GlcNAc-1-phospho-transferase that phosphorylates mannose residues in glycoproteins sorted to lysosomes (Table 4) [107]. The last identified substrate is the (pro)renin receptor which is shed by SKI-1/S1P in the bloodstream where it plays a major role in hypertension [108]. In humans, *MBTPS1* mutations are linked to the Silver–Russell syndrome (SRS) in patients affected by defective inter-organelle protein trafficking and skeletal malformations [109,110].

The unique cleavage specificity of SKI-1/S1P puts it in a class of its own, distinct from the basic-aa specific convertases. This led to the search for possible viral glycoprotein substrates that are activated via a non-canonical Furin(-like) cleavage, possibly by SKI-1/S1P or through a combination of both processing enzymes. Thus, SKI-1/S1P was shown to play a major role in the processing of surface glycoproteins precursors (GP-C) of infectious arenaviruses such as Lassa virus (LASV) at **R**R**L**L↓GT (Table 4) [111,112,113], which is endemic in West Africa and is estimated to affect some 500,000 people annually resulting in several thousand deaths each year [114]. 

Interestingly, while SKI-1/S1P is predicted to be fully mature in the *cis*/*medial* Golgi, it can cleave the GP-C of LASV in the ER/*cis* Golgi indicating a functional activity in early secretory compartments. We speculate that forms of SKI-1/S1P that are only cleaved at the B’/B sites can already be partially active. This may be due to the fact that the optimal **R**R**L**L↓ cleavage motif [14] found in site C of the SKI-1/S1P prodomain (Table 4) is identical to the processing site of LASV GP-C [111,112], representing an ideal substrate [14]. In line with this hypothesis, the Stefan Kunz group found that Golgi-cleavable substrates can be processed in the ER when engineered to contain **R**R**L**L↓, regardless of the rest of the backbone protein that carries the motif [112]. The N-terminal B’/B fragment of the prodomain represents an autonomous structural and functional unit that is necessary and sufficient for folding and partial activation of SKI-1/S1P. In contrast, the C-terminal B-C fragment lacks a defined structure but is crucial for autocatalytic processing and full enzymatic activity [115]. Therefore, it is not surprising that the partial processing of the prodomain in the ER at the B’/B site is enough to confer some enzymatic activity on specific viruses with very favorable glycoprotein processing sites, e.g., LASV. 

Attempts to generate vaccines against the GP-C of LASV revealed that major histocompatibility class I (MHC-I) receptors play a critical role in the T-cells response to LASV infections [116], suggesting that conditions that favor higher levels of MHC-I receptors may enhance the inflammatory response by activated macrophages following LASV infection contributing to its pathogenesis. In this context, the recent observation that PCSK9 inhibition/silencing in vivo enhances the levels of MHC-I via its PCSK9-induced degradation in endosomes/lysosomes [19] would suggest that this clinically safe treatment, though very effective in reducing hypercholesterolemia [117] and in cancer/metastasis treatments [19], may actually also enhance the contributions of the immune response to viral disease severity by exacerbating the LASV-induced inflammatory response, which represents a major hurdle in vaccine development against viral infections [118]. Thus, administration of PCSK9 inhibitors [117] would not be recommended in LASV-infected and HCV co-infected patients, where MHC-I and LDLR, respectively, play important roles in the pathogenesis. 

Another GP-C activated by SKI-1/S1P includes the prototypic arenavirus lymphocytic choriomeningitis virus (LCMV) processed at **R**R**L**A↓GT (Table 4) [112,119,120,121]. In contrast to LASV, SKI-1/S1P cleaves the GP-C of LCMV in the *trans* Golgi. Of note, the **R**R**L**A↓ motif is sufficient to direct the maturation of the glycoprotein within this late compartment. Indeed, a single point mutation of LASV GP-C engineered to replace Leu at P1 with Ala results in a glycoprotein resistant to the cleavage in the ER but sensitive to SKI-1/S1P activity much later in the secretory pathway [112,122] (Figure 1). These key findings by Stefan Kunz are important to further classify SKI-1/S1P as a unique enzyme, different from the rest of the basic aa-specific PC-members. Using the Arenaviruses as a tool, his team was able to confirm not only that SKI-1/S1P possesses a distinctive *consensus* motif but also to show that the nature of the single amino acids around the cleavage site is critical to allow the enzymatic activity within a specific sub-cellular compartment. Such behavior in the panorama of proteases is unique and probably reflects the complex maturation mechanism of SKI-1/S1P that Stefan Kunz unraveled, resulting in a plethora of forms retaining prodomain fragments of this enzyme. 

LASV and LCMV were the first Arenaviruses found to be SKI-1/S1P dependent [111]. Since then, other members of the Old World (OW) Arenaviruses showed this distinctive feature. The GP-C of Lujo Virus (LUJV) recently isolated in a cluster of lethal hemorrhagic fever cases in Africa, is an example [123]. Using a very elegant approach, Stefan Kunz showed that SKI-1/S1P processes the LUJV GP-C by engineering a luciferase sensor carrying the glycoprotein specific cleavage site **R**K**L**M↓ [124]. Like Old World counterparts, the New World (NW) arenaviruses—which are endemic in the Americas—strictly depend on SKI-1/S1P for their maturation.

The JUNV transmitted by rodents is a well characterized member of the NW arenaviruses with a mortality rate of 20–30% among infected populations in Argentina [125]. The JUNV surface glycoprotein GP-C is cleaved by SKI-1/S1P at **R**S**L**K↓AF. The processing site was confirmed by Stefan Kunz using the luciferase sensor platform [124] (Table 4). Interestingly, the JUNV maturation site closely resembles the C-site auto-cleavage motif of SKI-1/S1P [11] (Table 4). As expected, it was suggested that the JUNV envelope glycoprotein is activated in a late cellular Golgi compartment [126] where also the last steps of SKI-1/S1P maturation take place. In general, the present data suggest that arenavirus GP-Cs evolved to mimic SKI-1/S1P auto-processing sites [127,128], likely ensuring effective cleavage and avoiding competition with cellular substrates of SKI-1/S1P [129]. Old World and clade C New World GP-Cs further possess a distinctive signature that consists in displaying an aromatic residue at position P7, which is far away from the actual cleavage site. Such aromatic residues are implicated in the molecular recognition of the counterpart Tyr_285_ located on the molecular surface of the enzyme. This finding suggests that during co-evolution with their mammalian hosts, arenavirus GPCs expanded the molecular contacts with SKI-1/S1P beyond the classical four-amino-acid recognition sequences and currently occupy an extended binding pocket [113]. The specificity of the interaction between Tyr_285_ of SKI-1/S1P and aromatic P7 residues found in viral but not cellular substrates makes this interaction a promising target for the development of specific antiviral drugs which should not interfere with the catalytic triad directly, thus preserving full cellular activities of SKI-1/S1P. Overall, Stefan Kunz work has brought to light the outstanding ability of Arenaviruses to exploit SKI-1/S1P in a way to optimize the molecular contacts, outmatching the enzyme natural substrates. His results should be of great value for further identification of antivirals.

The crucial role of SKI-1/S1P in the virus life cycle and therefore in the establishment of infection has been beautifully shown by a recent work by Manning and colleagues. Candid #1 (Can), derived from the pathogenic Junin virus, is a vaccine against JUNV whose use is limited to Argentina. One single mutation of the envelope glycoprotein of Candid #1 is primarily responsible for the virus attenuation by impairing an effective GP-C glycosylation that in turn drastically limits the envelope glycoprotein processing [130]. The above example relies on a variation that perturbs the recognition of the viral substrate by the cellular enzyme. In a similar fashion, we can imagine that misfunctions of SKI-1/S1P can lead to similar outcomes. Indeed, studies on mice carrying a hypomorphic mutation in the *MBTPS1* gene (woodrat) showed that arenaviruses are incapable of establishing persistent infections in vivo [131]. Besides natural (Candid #1) and engineered (woodrat) variants, pharmacological treatments targeting SKI-1/S1P were also shown to be effective against arenavirus spread. Different from other enveloped viruses, only the cleaved glycoprotein is incorporated into budding virions. Naked viral particles fully lacking spikes are not infectious and cannot be primed by other proteases, e.g., during entry. Thus, SKI-1/S1P mediated maturation of arenavirus glycoproteins represents a key event and a druggable target, prompting several groups to test inhibitors against various arenaviruses [132]. 

SKI-1/S1P inhibitors can be grouped into three major classes: Proteins: We first showed that overexpression of the prodomain alone of SKI-1/S1P, as well as variants of α1-antitrypsin, can inhibit the function of this convertase [13]. The effect is striking against LASV multicycle replication [133]. Peptides: These include among others, tetrapeptide-chloromethylketones (CMKs) developed in our group [14] and the enediynyl peptides [134]. CMKs potently block LCMV infection [135] but their use is limited to research purposes since they are highly toxic in vivo.Non-peptide small molecules: Because of their properties and stability, this class of inhibitors are generally preferred over others for in vivo use. A small molecule SKI-1/S1P inhibitor PF-429242 was developed by Pfizer [136,137] and tested as an antiviral targeting GP-C processing and productive infection of arenaviruses. SKI-1/S1P inhibition by PF-429242 suppresses viral replication in cells infected with LASV, LCMV [138], and New World arenaviruses [139]. Interruption of drug treatment did not result in re-emergence of infection, indicating that PF-429242 treatment leads to virus extinction. Of note, Stefan Kunz found that the drug is capable of clearing LCMV from chronically infected cells with no emergence of escape variants [139]. This finding is intriguing since an LCMV mutant engineered to carry the RR**RR**↓ mutation is viable and fit [135]. The replacement of the wild type RRLA↓ motif with RR**RR**↓ does switch LCMV dependence from SKI-1/S1P to Furin. Therefore, it seems that arenaviruses are not prompted to use other members of the PCs family. The reason(s) for this selectivity of SKI-1/S1P has not yet been elucidated.

SKI-1/S1P also cleaves the glycoprotein PreGn of the Crimean Congo Hemorrhagic Fever Virus (CCHFV), a member of the *Bunyaviridae* family, at **R**R**L**L↓SE [140] (Table 4) to release an N-terminal mucin-like domain (MLD) fused to a non-structural glycoprotein GP38 and a C-terminal fusogenic Gn-containing domain. This virus is unique since after the SKI-1/S1P processing at **R**R**L**L↓, the N-terminal product is further processed at **R**S**KR**↓ by Furin(-like) enzymes separating the MLD from GP38 [141,142]. Functional data further revealed that cleavage at the Furin(-like) **R**S**KR**↓ site is not essential for CCHFV production or cell-to-cell spread, but that Furin cleavage enhances virion production [142]. In contrast, processing at **R**R**L**L↓ is critical for virus infectivity. Cells deficient in SKI-1/S1P produce no infectious virus, although PreGn accumulates normally in the Golgi apparatus where the virus assembles [141].

Aside from direct glycoprotein cleavage by SKI-1/S1P, indirect roles in viral infection are now being documented. In a first example, while SKI-1/S1P is not implicated in the direct processing of the surface glycoprotein GP of the Ebola virus (EBOV), the SKI-1/S1P activation of α/β-GlcNAc-1-phospho-transferase is required to allow cathepsins sorting to lysosomes and their promotion of EBOV assembly [143]. Thus, active cathepsins B and L in lysosomes cleave EBOV GP to remove the glycan cap and the mucin-like domains, thereby favoring binding of GP to the NPC1 receptor and priming it for fusion [143]. 

In a second example, loss of SKI-1/S1P activity is expected to reduce the processing of SREBP1,2 and hence reduce the levels of intracellular lipids that in turn drastically impact viral infections and replications. In line with this vision, the SREBP-dependent lipidomic reprogramming represents an attractive antiviral target [144]. However, since lipids are at the crossroad of various cellular pathways/signaling, deciphering the complex mechanism(s) behind the SREBPs-mediated antiviral effects may plead different explanations. Accordingly, robust inhibitory effects on SARS-CoV-2 entry and replication are observed following *MBTPS1* silencing or inhibition of SKI-1/S1P by PF-429242 treatment (IC_5_0 = 300 nM) in cells [145]. The present evidence excludes any interference with the maturation of the envelope S-glycoprotein, which is indeed cleaved by another Proprotein Convertase, i.e., Furin [74]. Rather, this member of the *Coronaviridae* family depends on healthy cellular lipid homeostasis more than other coronaviruses through a mechanism that has not yet been fully described. Lipid droplets formation were further confirmed to be essential in SARS-CoV-2 replication and pathogenesis [146]. On the other hand, it was suggested that inhibition of SREBP2 activation (implicating SKI-1/S1P cleavage) may represent a successful strategy to avoid a possible cytokine storm event in COVID-19 patients [147,148]. Flaviviruses represent another group of pathogens whose ability to infect is intimately associated with cholesterol and therefore SKI-1/S1P. Along this line, flaviviruses rearrange intracellular membranes and orchestrate a profound reorganization of the host cell lipid metabolism to create a favorable environment for viral replication and assembly. Major members of the *Flaviviridae* family are the Dengue virus (DENV), hepatitis C virus (HCV), West Nile virus (WNV), and Zika virus (ZIKV). DENV is highly sensitive to SKI-1/S1P inhibitors at the late stage of the infection indicating that PF-429242 does not target virus surface glycoprotein prM processing directly, but rather regulates host cellular factors needed to synthesize lipids which are essential for suitable assembly platforms [149]. The inhibitory mechanism may be associated with a dramatic reduction in the abundance of neutral lipids and their markers which drastically impairs virus propagation. Similarly, we [150] and others [151] showed that mis-regulation of the cellular SKI-1/S1P activity is a successful strategy to eradicate HCV infection.

WNV and ZIKV were likewise shown to be sensitive to SKI/S1P inhibitors, along with structurally unrelated drugs targeting the SREBPs pathway [152]. Interestingly, we showed that SKI-1/S1P regulates HCV viral replication early in the viral lifecycle with no change in LDLR or CD81 receptor levels [150], closely resembling the effect of SKI-1/S1P inhibition in infections by SARS-CoV-2 [148]. In this scenario, indirect effects induced by SKI-1/S1P inhibition may contribute to reducing virus infection [153]. 

It is plausible that in the presence of PF-429242, lower cholesterol and fatty acids syntheses may upregulate the antiviral interferon (IFN) response of cells, as it was observed by PCSK9 inhibitors that would lower ER-cholesterol availability and hence active SREBP-2 levels, resulting—for example—in reduced DENV assembly [103]. Flaviviruses are somehow expected to be affected by SKI-1/S1P manipulation due to their lipid dependence during virion assembly and/or viral protein trafficking. In contrast, there are pathogens that are mainly affected in their entry step upon disrupting the normal cellular lipid distribution. This is the case of Hantaviruses, members of the family *Bunyaviridae* associated with hemorrhagic fever with renal and cardiopulmonary/pulmonary syndromes in humans. Both depletion of *MBTPS1* gene and SKI-1/S1P inactivation efficiently inhibit Hantavirus membrane fusion at or near the lipid mixing step and block cytoplasmic delivery of viral matrix protein entry. As a matter of fact, Hantavirus membrane fusion directly depends on the cholesterol abundance of the target cells [154]. 

It is becoming more and more evident that SKI-1/S1P is a key regulator of multiple types of viral infection. Thus, targeting such a host component represents a powerful strategy to contain the multiplication and spread of viruses. In turn, this would provide the patient’s immune system a window of opportunity to develop an appropriate antiviral immune response to clear the virus off. Unfortunately, the only small inhibitor available PF-429242 failed to reach clinical applications due to its poor pharmacokinetic properties [137], thus begging for a potent and safe SKI-1/S1P inhibitor as an antiviral to fight such deadly infections [132,155,156,157].

## 3. Discussion

From the above considerations, it became clear that most enveloped viruses require the processing of their surface glycoproteins for productive infectivity. Enhanced pathogenicity has been associated with the apparition of a new cleavage site in certain viruses. While clearly favoring Furin cleavage [74,158], the exact role of the acquisition of a Furin site in SARS-CoV-2 [70] and its role in the spread of the virus and/or its pathogenicity have not been adequately established. Experimental evidence is not easy to obtain. One would need to generate a virus mutant to be tested in an animal model exhibiting the multiple pathogeneses that mimic the human clinical manifestation of COVID-19. A clear advantage of SARS-CoV-2 over SARS-CoV-1 has been its ability to spread asymptomatically and with great efficiency in a large portion of the population. One can speculate that the acquisition of a Furin cleavage site has contributed to confer to SARS-CoV-2 such an advantage and to enhance the tropism of this virus, as Furin is widely expressed in most tissues, whereas TMPRSS2 implicated in SARS-CoV-1 activation [24] is more limited in its tissue expression.

A clear distinction was made between the choice of the two ubiquitously expressed convertases: (i) the basic aa-specific Furin *versus* (ii) the non-basic aa-specific SKI-1/S1P. The mechanistic rationale may be, in part, related to the subcellular localization of the active forms of these convertases, which are initially synthesized as zymogens in the ER that are autocatalytically activated via two or more cleavages of their prodomains. Thus, whereas SKI-1/S1P autocatalytically activates itself early in the ER/*cis* Golgi (B’/B cleavages), becoming maximally active in the *medial* Golgi (C/C’ cleavages), Furin is first autocatalytically processed at the C-terminus of its prodomain in the ER, but still remains inactive as it is non-covalently bound to its prodomain. It is only in the TGN that Furin completely separates from its inhibitory prodomain, whereupon it processes various substrates *in trans* in the TGN, cell surface and/or endosomes [9,20,159,160]. This allows the processing of a wide spectrum of substrates that are preferentially cleaved by Furin at either the cell surface neutral pH (Anthrax toxin and PCSK9) or acidic pH of the TGN (e.g., SARS-CoV-2 S-protein, Ebola Zaire pro-GP) and endosomes (e.g., Diphtheria and Shiga toxins). The same also seems to apply for SKI-1/S1P that can be partially activated at neutral pH in the ER, or much more active in the slightly acidic pH environment of the *cis/medial* Golgi (Figure 2). An additional selectivity is provided by the substrate itself via its protein folding and subcellular trafficking, whereupon some SKI-1/S1P substrates are activated in the ER (e.g., LASV GP-C), and the majority of others in the *cis/medial* Golgi (SREBPs, and multiple viral glycoproteins). Finally, one single pathogen is known to exploit both pathways; CCHFV uses both SKI-1/S1P for viral activation and Furin to enhance viral production [142]. 

In all examples described above, the link between protease(s) and the pathogen is clear cut, being the surface glycoprotein matured into active forms (e.g., acquisition of fusogenic properties) to attain full infectivity. This event is visible at the cell surface and therefore easily traceable by standard biochemical techniques. In contrast, how pathogens interact with the host as well as hijack, avoid and re-programme pre-existing pathways is still a matter of study. We may identify some common strategies, keeping in mind—though—that each infectious agent leaves its own fingerprints behind. For instance, upregulation of the SKI-1/S1P-dependent SREBP1,2 pathway is a well-documented example of host manipulation by Flaviviruses, since the increase of ER/Golgi lipids is critical for virion assembly, as was recently shown for the Dengue virus [103]. Other viruses, e.g., SARS-CoV-2 and Hantaviruses, require cholesterol for effective entry, thereby affecting early stages of infection. Along the same line, we are beginning to appreciate the master role of PCSK9, as a key regulator of cholesterol content. Some other pathways are carefully avoided by the pathogen allowing them to “silently” colonize the host without raising any red flag. This is the case of LCMV chronic infections that do not trigger any cellular unfolded protein response (UPR) pathway, the latter known to be activated by SKI-1/S1P mediated cleavage of ATF6 [161]. Overall, however, our current knowledge just gives us a glimpse over a very complex scenario where actors play roles in a plot not yet fully unraveled. More and more evidence, for example, start unveiling the contribution of PCs to the immune response: the same pathways that regulate lipid homeostasis—thought to be manipulated to favor the establishment of successful infections—are now shown to be a “wake-up call” for immunity [19,162,163]. In view of this situation, more efforts are required to fill the gaps. The current SARS-CoV-2 pandemic further highlights that the scientific community is due for a shake-up. We need to invest more energies and resources to better understand pathogens and, of course, to develop effective drug treatments. In this context, PCs represent ideal drug targets. First, by being proteases, these molecules possess defined substrate-binding pockets, which can be exploited as binding sites for pharmaceutical inhibitors [164]. In addition, the interference with PCs activity can be quantitatively measured by wellestablished assays that allow rational lead optimization. Second, PCs are at the crossroad of multiple types of infections. Identification of powerful and safe PCs inhibitors may come in handy for current and newly arising pathogens, including SARS-CoV-2 and the emergence of a Furin-dependent influenza virus. The latter has been a matter of discussion during the last decade as a causative agent of a potential novel global pandemic [165,166]. Likewise, targeting PCs would provide a powerful tool to modulate lipids, which are a “common denominator” of many enveloped viruses, e.g., flaviviruses, α-viruses, coronaviruses, filoviruses, retroviruses and more (Table 1).

Thus, PCs-targeted antivirals provide great potential as general pandemic-preparedness tools. The advantage of targeting the host—i.e., PCs—consists in delivering a drug to a patient early in the infection and before the pandemic occurs and without knowing any details on the nature of the pathogen other than its dependence on these enzymes. Moreover, PCs fall into the “usual suspects” category, meaning they are popular and very often hijacked proteases and their involvement in future diseases is highly likely. In answer to those who claim toxic effects by PC-inhibition of critical host factors, we highlight that the drug administration should cover only a limited amount of time (few days), sufficiently to allow an appropriate immune response to the pathogen. Also, pharmaceutical companies manage to find drugs and dosing regimens against host proteins that people can tolerate. So, why should PCs antivirals be any different?

The present review clearly emphasized the seminal contribution of **Stefan Kunz** to our better understanding of the complex interplay between hemorrhagic fever viral surface glycoproteins folding and processing activation by SKI-1/S1P. The work of his team led the way towards a rational understanding of the structure-function of SKI-1/S1P, its zymogen activation steps, and their effects on the processing of various arenaviruses that rely on this enzyme for their activation. The combination of his contributions to the monumental literature on the activation of enveloped viruses provided a distinct framework for future studies on the development of specific inhibitors or therapies against a multitude of infectious viruses. This is a solid base to build upon for our preparedness to counter the onslaught of the present (e.g., COVID-19) and future pandemics that will surely arise because of the unrelenting interference and manipulation of wildlife and fauna by man and the ensuing loss of biodiversity.

Finally, two of us (AP and NGS) have had the privilege to work closely with Stefan Kunz, allowing us to appreciate his scientific intellect and curiosity. He will be fondly missed and remembered, and this review is a tribute and dedication to his memory.

## Figures and Tables

**Figure 1 viruses-13-01229-f001:**
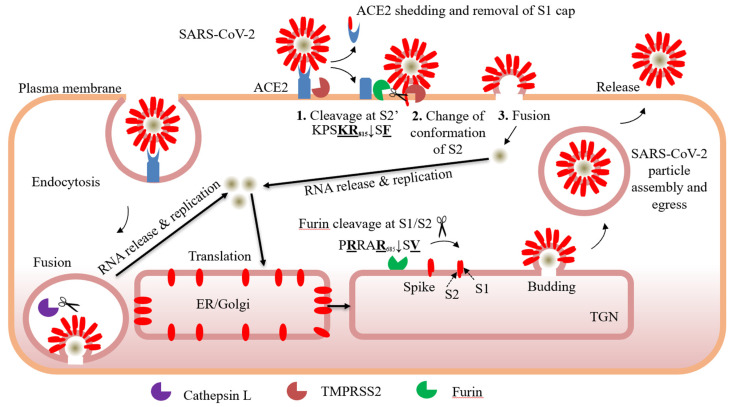
Role of protease processing in SARS-CoV-2 entry. SARS-CoV-2 can enter through fusion at the plasma membrane or by endocytosis. The processing of the spike-protein at the plasma membrane by Furin at the S1/S2 site is favored while Furin & TMPRSS2 are involved in the processing at S2′, thereby inducing fusion. Furin cleavage at S1/S2 is not required to promote entry through endocytosis and Cathepsin L may cleave at S1/S2 and S2′ in endosomes.

**Figure 2 viruses-13-01229-f002:**
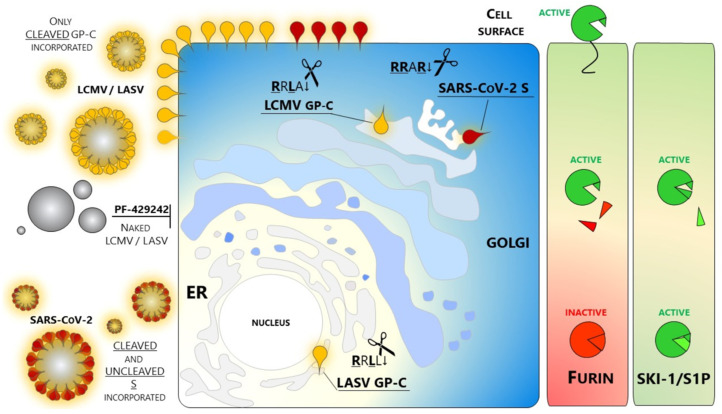
Schematic representation of the cellular compartments and where Proprotein Convertases are active against viral substrates. Furin processes the coronavirus SARS-CoV-2 spike S in the TGN; SKI-1/S1P matures the arenavirus LASV and LCMV GP-Cs in the ER and late Golgi, respectively. The cleaved viral glycoproteins reach the cell surface. Arenaviruses do incorporate only cleaved GP-C into viral budding particles since treatment with the SKI-1/S1P inhibitor PF-429242 blocks GP-C processing and its incorporation into budding particles; SARS-CoV-2 can include both cleaved and uncleaved envelope glycoprotein spike S.

**Table 1 viruses-13-01229-t001:** Families of pathogenic viruses that depend on the basic aa-specific proprotein convertases for host cell entry. RT = reverse transcriptase. Modified from [21].

Family	Virus	Capsid	Genome
*Retroviridae*	HIV, Leukemia viruses	Enveloped	Linear ssRNA(−), RT
*Flaviridae*	HCV, Dengue, Zika, West Nile	Enveloped	Linear ssRNA(+)
*Togaviridae*	Chikungunya	Enveloped	Linear ssRNA(+)
*Coronaviridae*	SARS-CoV-1,2, MERS	Enveloped	Linear ssRNA(+)
*Filoviridae*	Ebola, Marburg	Enveloped	Linear ssRNA(−)
*Orthomyxoviridae*	Avian Influenza H5N1	Enveloped	Linear ssRNA(−)
*Paramixoviridae*	Measle, RSV, Nipah, MPV	Enveloped	Linear ssRNA(−)
*Hepadnaviridae*	Hepatitis B	Enveloped	Linear ssDNA (−), RT
*Herpesviridae*	Herpes, CMV, Varicella-Zoster	Enveloped	Linear dsDNA
*Papillomaviridae*	HPV	Naked	Circular dsDNA

**Table 2 viruses-13-01229-t002:** Various enveloped viruses and the sequences of surrounding their surface glycoprotein cleavage sites (designated by an arrow) by Furin(-like) proprotein convertases. The bold and underlined residues at positions P8, P6, P4, P2, P1 and P2′ emphasize the importance of these amino acids for protease recognition.

Virus	Glycoprotein	P8		P6		P4		P2	↓		P2′
HIV	gp160			V	Q	**R**	E	**K**	**R**	A	**V**
H7N1 A/FPV/Rostock/34	HA			**K**	K	**R**	E	**K**	**R**	G	**L**
Avian H5N8 TKY/IRE	HA			**R**	K	**R**	K	**K**	**R**	G	**L**
Avian H5N1 A/HK/97	HA	**R**	E	**R**	R	**R**	K	**K**	**R**	G	**L**
Avian H5N1 TKY/ENG	HA	N	T	P	Q	**R**	K	**K**	**R**	G	**L**
Human CMV	gB			**H**	N	**R**	T	**K**	**R**	S	T
Human MPV	F Protein			N	P	**R**	Q	S	**R**	F	**V**
Human RSV	F Protein			**K**	K	**R**	K	**R**	**R**	F	**L**
Dengue Virus (DENG2)	PrM			**H**	R	**R**	E	**K**	**R**	S	**V**
Ebola Virus	gp160			G	R	**R**	T	**R**	**R**	E	A
Chikungunya (CHIKV)	E3E2			P	R	**R**	Q	**R**	**R**	S	**I**
Zika Virus	PrM			A	R	**R**	S	**R**	**R**	A	**V**
SARS-CoV-2	S			S	P	**R**	R	A	**R**	S	**V**

**Table 3 viruses-13-01229-t003:** Amino acid changes in the S-protein of SARS-CoV-2 variants of worldwide concern covering the period of September 2020–June 2021. The **P681H** and P681R emphasize their P5 position in the Furin(-like) site (P/H/R)RRAR_685_↓SV and variants in red are expected to affect the binding of S-protein to ACE2.

Variant	First Identification	S-Protein Mutations
B.1.1.7 α-variant	UK September 2020	del69-70 HV, del144Y, N501Y, A570D, D614G, **P681H**, T761I, S982A, D1118H
B.1.351 β-variant	South Africa October 2020	K417N, E484K, N501Y, D614G, A701V
B.1.1.248 γ-variant	Brazil, Japan January 2021	L18F, T20N, P26S, D138Y, R190S, K417T, E484K, N501Y, H655Y, T1027I
B.1.167 δ-variant	India December 2020	T95I, G142D, E154K, K417N, L452R, E484Q, D614G, **P681R**

**Table 4 viruses-13-01229-t004:** SKI-1/S1P processing sites of various cellular substrates and those of enveloped viruses with emphasis on the sequences surrounding their surface glycoprotein cleavage sites, designated by an arrow. The bold and underlined residues at positions P4 and P2 emphasize the importance of these amino acids for protease recognition.

	Substrate	P8		P6		P4		P2	↓		P2′		P4′
**CELLULAR**	h Pro-SKI-1 site B	R	K	V	F	**R**	S	**L**	K	Y	A	E	S
	h Pro-SKI-1 site B’	V	T	P	Q	**R**	K	**V**	F	R	S	L	K
	h Pro-SKI-1 site C	R	H	S	S	**R**	R	**L**	L	R	A	I	P
	h SREBP2	S	G	S	G	**R**	S	**V**	L	S	F	E	S
	h SREBP1	H	S	P	G	**R**	N	**V**	L	G	T	E	S
	h ATF6	A	N	Q	R	**R**	H	**L**	L	G	F	S	A
	h Luman	G	V	L	S	**R**	Q	**L**	R	A	L	P	S
	m OASIS (CREB3L1)	Q	M	P	S	**R**	S	**L**	L	F	Y	D	D
	h CREB-H	R	V	F	S	**R**	T	**L**	H	N	D	A	A
	h pro-BDNF	K	A	G	S	**R**	G	**L**	T	S	L	A	D
	h α/β-GlcNAc-1-pTr	K	N	T	G	**R**	Q	**L**	K	D	T	F	A
	h FAM20C	K	H	T	L	**R**	I	**L**	Q	D	F	S	S
	h pro-Renin receptor	I	R	K	T	**R**	T	**I**	L	E	A	K	Q
**VIRAL**	Lassa Virus (LASV) GP-C	I	Y	I	S	**R**	R	**L**	L	G	T	F	T
	CCHFV PreGn	S	S	G	S	**R**	R	**L**	L	S	E	E	S
	LCMV GP-C	K	F	L	T	**R**	R	**L**	A	G	T	F	T
	Junin Virus (JUNV) GP-C	Q	L	P	R	**R**	S	**L**	K	A	F	F	S

## Data Availability

Not applicable.

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
