# Peer review of "How Do Enveloped Viruses Exploit the Secretory Proprotein Convertases to Regulate Infectivity and Spread?"

_viruses, 2021, doi:10.3390/v13071229_

Round 1

Reviewer 1 Report

N. Seidah and co-authors present a comprehensive and very extensive review on the role of Furin-like PCs in the processing of entry related viral glycoproteins. The work is of high interest for the field and I have no objections to the publication of this work.

Some minor comments:

  • please do thorough spell checking, there were severa minor grammatical errors in the manuscript
  • the " ' " symbol is used to depict several different things, like secondary cleavage sites, secondary cleavage products etc, which is confusing. The authors should check again and use unique identifiers for the different purposes, e.g.: line 299-305 ' is used to depict cleavage fragments of proteins;  line 272 S2' is used to depict a secondary proteolytic site)
  • line 324, check if it is appropriate to include other references that study Hydroxychloroquine in non-human primate models to elucidate more the unclear situation regarding clinical benefits/anti-viral activity in vivo (e.g. Maisonnasse et al Nature 2020?)

Author Response

We thank the reviewer for his/her constructive criticisms. Below is a point by point detail of our modifications:

  1. We performed a thorough spell checking and corrected some errors.
  2. We modified the sentences to clearly explain the difference between S2' product and S2' site as follows:

The S2-product generated following S1/S2 cleavage contains a second proteolytic site (called S2’ cleavage site), which when cleaved would generate an S2’-fragment that starts with a fusion peptide (FP) followed by two heptad-repeat domains preceding the transmembrane domain (TM) and cytosolic tail. Cleavage at the S2’ site triggers membrane fusion and is essential for efficient viral infection. The S2’ cleavage sites of SARS-CoV-1 (PTKR797↓SF) and MERS-CoV (RSAR887↓SA) suggest that both could be cleaved by a Furin(-like) enzyme, but the single Arg-specific enzyme TMPRSS2 has solely been proposed to cleave both spike glycoproteins at the S2’ site [69].   

The Furin(-like) S2’ cleavage site at KR797↓SF with P1 and P2 basic residues and a P2’ hydrophobic Phe [9], is identical between the SARS-CoV-1 and SARS-CoV-2 (Fig. 1). In the MERS-CoV and OC43-CoV it is replaced by the less favourable Furin(-like) site RXXR↓SA, with P1 and P4 basic residues, and an Ala (not hydrophobic) at P2’. However, in other less pathogenic circulating human coronaviruses, the S2’ cleavage site only exhibits a Furin-unfavourable monobasic R↓S sequence [70] with no basic residues at either P2 and/or P4. Even though processing at the S2’ site in the spike glycoprotein of SARS-CoV-2 is thought to be key in the activation of the S-protein, leading to cell-fusion and entry, multiple protease(s) might be involved in S-cleavage at different sites and subcellular compartments [71]. The ability of the Arg/Lys-specific TMPRSS2 to directly cleave at the S2’ site was inferred from the viral entry blockade by the relatively non-specific TMPRSS2 inhibitor Camostat [72, 73]. We recently demonstrated that Furin is capable of performing both S1/S2 (PRRAR685↓SV) (Table 2) and S2’ (KPSKR815↓SF) cleavages generating an N-terminal S1 subunit (with an RBD domain) and a C-terminal membrane bound fusogenic S2’-fragment in the presence of the Spike-glycoprotein receptor ACE2 [74]. Interestingly, binding of the RBD of the S-protein of SARS-CoV-2 to ACE2 was demonstrated to exert a conformational change that allosterically enhances the exposure of the S1/S2 site to Furin cleavage [75]. Whether it also enhances exposure of the S2’ site is not clear, but our results clearly show that Furin cleavage at the S2’ site is enhanced in the presence of ACE2 [74]. Our data also showed that non-peptide small molecule BOS-inhibitors of Furin(-like) enzymes block the processing of the S-protein at both S1/S2 and S2’ sites and result in a significant reduction of cell-to-cell fusion. In the presence of overexpressed TMPRSS2 together with the S-protein, the latter was cleaved into an apparently lower sized fragment than S1 (called S1’) [74] released in the cell culture media, which turned out to be an authentic S1 product lacking O-glycosylation as it is generated in the endoplasmic reticulum (ER) and secreted by an unconventional pathway (submitted). Furthermore, overexpressed TMPRSS2 generated two C-terminal membrane bound fragments S2a and S2b, which are S2 and S2’ products lacking O-glycosylation. These products may be artificially produced in the ER upon overexpression of TMPRSS2 with the S-glycoprotein. However, in co-culture conditions in which ACE2 and TMPRSS2 are expressed in acceptor cells and S-protein in donor cells we concluded that both Furin (in the TGN) and TMPRSS2 (at the cell surface) cleave S-protein into S1/S2 and S2’ in the presence of ACE2 (submitted), but that in addition TMPRSS2 can also process and shed ACE2 into a soluble form thereby modulating viral entry and infection.  In agreement, an inhibitor cocktail that combines a Furin inhibitor (BOS) with a TMPRSS2 inhibitor (Camostat) can reduce by 99% SARS-CoV-2 infectivity of lung-derived Calu-3 cells [74]. It should be noted that coronaviruses may enter the cells through fusion at the plasma membrane or following endocytosis into endosomes [76]. The versatility of strategies to penetrate the cell allow coronaviruses to depend less on the availability of the different proteases in specific cell type and in cellular compartments. Furthermore, the activation process of SARS-CoV-2 may be more complex, as other proteases may also participate in processing the S-protein at multiple sites  in various tissues, including cathepsins [71], HAT [77], TMPRSS11D and TMPRSS13 [78].

3. We have added the reference of Maisonnasse et al. (new ref # 81).

Reviewer 2 Report

The review „How do enveloped viruses exploit the secretory proprotein convertases to regulate infectivity and virus spread?“ by Seidah and colleagues summarizes the knowledge of the role of proprotein convertases in the activation of viral surface glycoproteins.

The paper starts with an historic overview of past and current pandemics. After a short introduction of the proprotein convertase family, the role of furin in viral replication as the most prominent proprotein convertase is further discussed. Furin is involved in the cleavage of glycoproteins of various viruses like highly pathogenic avian influenza viruses, Chikungunya virus or Zika virus. Recent findings also indicate that furin plays a role in the activation of the spike protein of coronaviruses. This part is discussed in the second part of the paper. The third part is about the function of PCSK9 in hepatitis C and dengue virus infection. In the last part of this review the role of SKI-1/S1P in viral infections is discussed. SKI-1/S1P differs from the other proprotein convertases because it recognizes a different non-basic cleavage pattern. It plays a crucial role in the replication cycle of arenaviruses. Taken together, proprotein convertase inhibitors represent a promising therapeutic approach to treat various viral infections.

There are several reviews on the role of proprotein convertases in diseases, amongst others the reviews from Artenstein et al. or Braun et al.. The review from Izaguirre also focusses on the role of proprotein convertases on virus cell entry. However, most of these articles focus on the basic proprotein convertase furin. The review of Seidah and colleagues provides an interesting overview not only of furin but also on the knowledge of the role of SKI-1/S1P and PCSK9 in viral infections. Furthermore, the authors comment on the latest results of furin’s role in coronavirus replication with a focus on SARS-CoV-2.

Comments:

  • Several abbreviations are not introduced. I would recommend to add a list of abbreviations.
  • Table 1 looks very similar to Table 1 from the Izaguirre review (Viruses2019,11, 837; doi:10.3390/v11090837). This review is later cited, but I would recommend to cite it in the legend or to revise the table.
  • I would recommend to write „Furin(-like) cleavage site“ instead of „Furin-like sequence/site“, which is a bit misleading.
  • The sentence on page 4 lines 129-131 needs clarification. Fowl plague virus is used as the abbreviation for H7N1 (A/FPV/Rostock/34), please add the strain designation. Furthermore, the original paper from Stieneke-Gröber et al. (EMBO J. 1992 Jul; 11(7): 2407–2414) should be cited here. Additionally, I think H5N1 is missing in the sentence because FPV is also an influenza virus.
  • Please use a different colour code for the amino acids in Table 2. Both the two yellow colours as well as the two orange colours are not easy to distinguish.
  • The activation process of coronaviruses is complex and not fully understood. Also other proteases like HAT, DESC-1 or TMPRSS13 have been shown to activate the spike proteins of SARS- and MERS-CoV.
  • Please give an explanation for the colour coding or bold/underlined font in Table 3.

Author Response

We would first like to thank this reviewer for his/her constructive criticisms. Below we detail our answers:

  1. We have now added a list of abbreviations.
  2. We have referred to the review of Izaguirre in the legend in addition to the one referred to in the text (reference  21 ). We stated in the legend that the table is a modified form of the one presented in ref. 21.
  3.  As recommended, we replaced all Furin-like sites by Furin(-like) sites.
  4.  We modified Table 2 to include H7N1 A/FPV/Rostock/34 and corrected H5 to H5N1. 
    In the text, we wrote: In 1992, Furin was identified to be the cellular protease cleaving hemagglutinin (HA) of fowl plague/influenza virus (FPV; H7N1 A/FPV/Rostock/34), as well as the HIV glycoprotein gp160 into gp120 and gp41 [28-30]. This included the original reference of Stieneke-Gröber et al. in 1992 (now NEW reference 30).

  5.  The colors in Table 2 have been modified to make them more visible, as recommended.
  6. At the end of the SARS-CoV-2 third paragraph, we have added the following sentence:

    Furthermore, the activation process of SARS-CoV-2 may be more complex, as other proteases may also participate in processing the S-protein at multiple sites  in various tissues, including cathepsins [71], HAT [77], TMPRSS11D and TMPRSS13 [78].

  7. We simplified Table 3 and provided in the legend an explanation for the colors as follows:
    Table 3. Amino acid changes in the S-protein of SARS-CoV-2 variants of worldwide concern covering the period of September 2020–January 2021. The P681H emphasizes its P5 position in the Furin(-like) site (P/H)RRAR685↓SV and variants in red are expected to affect the binding of S-protein to ACE2.